# The Potential of *Phaeodactylum* as a Natural Source of Antioxidants for Fish Oil Stabilization

**DOI:** 10.3390/foods11101461

**Published:** 2022-05-18

**Authors:** Robbe Demets, Simon Van Broekhoven, Lore Gheysen, Ann Van Loey, Imogen Foubert

**Affiliations:** 1Research Unit Food & Lipids, Campus KULAK, KU Leuven, E. Sabbelaan 53, 8500 Kortrijk, Belgium; robbe.demets@kuleuven.be (R.D.); simon.vanbroekhoven@student.kuleuven.be (S.V.B.); lore.gheysen@kuleuven.be (L.G.); 2Leuven Food Science and Nutrition Research Centre (LFoRCe), KU Leuven, Kasteelpark Arenberg 20, 3001 Leuven, Belgium; ann.vanloey@kuleuven.be; 3Centre for Food and Microbial Technology, Laboratory of Food Technology, KU Leuven, Kasteelpark Arenberg 22, 3001 Leuven, Belgium

**Keywords:** lipid oxidation, n-3 LC-PUFA, fucoxanthin, SPME arrow

## Abstract

Worldwide, fish oil is an important and rich source of the health-beneficial omega-3 long-chain polyunsaturated fatty acids (n-3 LC-PUFA). It is, however, troubled by its high susceptibility towards lipid oxidation. This can be prevented by the addition of (preferably natural) antioxidants. The current research investigates the potential of *Phaeodactylum* carotenoids in this regard. The oxidative stability of fish oil and fish oil with *Phaeodactylum* addition is evaluated by analyzing both primary (PV) and secondary (volatiles) oxidation products in an accelerated storage experiment (37 °C). A first experimental set-up shows that the addition of 2.5% (*w*/*w*) *Phaeodactylum* biomass is not capable of inhibiting oxidation. Although carotenoids from the *Phaeodactylum* biomass are measured in the fish oil phase, their presence does not suffice. In a second, more elucidating experimental set-up, fish oil is mixed in different proportions with a *Phaeodactylum* total lipid extract, and oxidative stability is again evaluated. It was shown that the amount of carotenoids relative to the n-3 LC-PUFA content determined oxidative stability. Systems with a fucoxanthin/n-3 LC-PUFA ratio ≥ 0.101 shows extreme oxidative stability, while systems with a fucoxanthin/n-3 LC-PUFA ratio ≤ 0.0078 are extremely oxidatively unstable. This explains why the *Phaeodactylum* biomass addition did not induce oxidative stability.

## 1. Introduction

Fish oil is worldwide an important source of omega-3 long-chain polyunsaturated fatty acids (n-3 LC-PUFA) for both food (supplements or food enrichment) and feed (aquaculture, cattle, and poultry) applications [1,2]. N-3 LC-PUFA are known for their strongly recognized health benefits [3]. Fish oil can contain up to 35% of n-3 LC-PUFA, making it a rich source of these fatty acids [4]. However, due to their unsaturated nature and these high amounts, fish oil is highly susceptible to lipid oxidation. As this chemical deterioration process causes a loss of nutritional value and the development of both rancid flavors and potentially toxic compounds, it has to be eliminated [5]. Consequently, strong synthetic antioxidants such as BHA, BHT, and TBHQ have successfully been applied, but given their potential negative health effects restricting their usage and a growing interest in natural ingredients, the search for natural antioxidants able to inhibit fish oil oxidation has received a lot of attention [6]. Apart from tocopherols and rosemary extracts being the most frequently used natural antioxidants, a broad range of plant sources containing polyphenolic compounds has been explored, as reviewed by Hrebień-Filisińska [7]. Studies on marine natural antioxidant sources for the inhibition of fish oil oxidation, on the other hand, have been rather limited with the major attention attributed to macroalgal extracts [8,9,10,11,12,13]. Microalgae have only scarcely been addressed, with Golmakani et al. [14] showing the potential of an *Arthrospira platensis* extract in stabilizing kilka oil. However, microalgae harbor a lot of antioxidative potentials [15]. Apart from being considered a novel source of n-3 LC-PUFA (being the primary producers of these fatty acids), certain photoautotrophic microalgal species possess high oxidative stability despite having a high content of n-3 LC-PUFA. Ryckebosch et al. [16] showed that n-3 LC-PUFA rich photoautotrophic microalgal oils are far more oxidatively stable than fish oil, and Gheysen et al. [17,18] demonstrated the oxidative stability of photoautotrophic n-3 LC-PUFA rich microalgal biomass in aqueous suspensions, attributing it to the presence of endogenous, antioxidative carotenoids. The question, therefore, arises whether photoautotrophic n-3 LC-PUFA rich microalgae could also serve as a source of natural antioxidants on top of being considered as an alternative/additional source of n-3 LC-PUFA. To the best of our knowledge, no former research has been performed on the potential of n-3 LC-PUFA rich microalgae for fish oil stabilization.

Research on microalgae as a source of antioxidants in bulk oil has up to now been focused on vegetable oils. Moreover, apart from Lee et al. [19] showing inhibition of soybean oil oxidation by the addition of *Nannochloropsis* biomass, all studies in this regard focused on non-‘n-3 LC-PUFA rich’ species to which either microalgal biomass or a microalgal extract is added. Studies with microalgal biomass addition encompassed the addition of either *Chlorella* or *Arthrospira* to olive oil, and all reported enhanced oxidative stability attributable to the gradual release of antioxidants [20,21,22,23,24]. Studies with microalgal extract addition were more diverse in terms of species (*Haematococcus* [25,26], *Scenedesmus* [27], *Chlorella* [28], and *Anabaena* [6]), extraction method (solvent extraction with methanol, hexane, or acetone or supercritical fluid extraction with CO_2_) and vegetable oil (soybean, sunflower, olive, or others) but also reported an increase in oxidative stability explainable by the presence of carotenoids. Only Gouveia et al. [28], investigating the addition of a *Chlorella* supercritical CO_2_ extract to soybean oil, did not observe an antioxidative effect.

Based on these studies, it is clear that microalgae show potential for the inhibition of bulk oil oxidation. The aim of this research was to investigate whether this also holds for fish oil as an n-3 LC-PUFA rich and therefore oxidatively more susceptible oil, using n-3 LC-PUFA rich microalgae known to be stable towards n-3 LC-PUFA oxidation. For this, *Phaeodactylum* sp. was selected. On a dry weight basis, this microalga contains around 20% of lipids [29], of which eicosapentaenoic acid (EPA) (an n-3 LC-PUFA), C16:0, C16:1, and C14:0 form the most important fatty acids [30]. It is also rich in carotenoids, with fucoxanthin being the major one next to diatoxanthin and β-carotene [18]. Foo et al. [31] have already suggested the antioxidative capacities of *Phaeodactylum*, although only based on fast antioxidant assays. The experimental set-up was twofold. In the first part, the antioxidative effect of supplementing fish oil with *Phaeodactylum* sp. biomass was investigated by analyzing both primary (PV by FOX) and secondary (volatiles by HS-SPME Arrow GC-MS) oxidation products throughout an accelerated storage experiment (3 weeks at 37 °C). In the second part, an elucidating approach was followed to obtain more insight into the required amount of microalgal antioxidants needed to inhibit n-3 LC-PUFA oxidation and to explain the results of the first experimental set-up. For this, a total lipid extract (chloroform-methanol 1:1) of the *Phaeodactylum* biomass was made and mixed with fish oil in different proportions to obtain six different systems (containing 100, 79, 49, 6, 1, and 0 weight% of the *Phaeodactylum* total lipid extract). Again, the oxidative stability was evaluated by analyzing primary and secondary oxidation products throughout an accelerated storage experiment (10 weeks at 37 °C). Other than the first experimental set-up, the second was not aimed at a practical application. Rather than investigating the potential of the addition of *Phaeodactylum* total lipid extract as an antioxidant, the objective was to clarify the n-3 LC-PUFA oxidation inhibition capacity of the antioxidative compounds present in this extract from a scientific point of view.

## 2. Materials and Methods

### 2.1. Materials

#### 2.1.1. Raw Materials

Freeze-dried *Phaeodactylum* sp. biomass was obtained from Archimede Ricerche (Camporosso, Italy). The n-3 LC-PUFA content was determined as 20.0 (±0.3) mg/g dry biomass and the carotenoid profile consisted of 5.31 (±0.52) mg fucoxanthin, 0.61 (±0.11) mg diatoxanthin and 0.48 (±0.02) mg β-carotene per g dry biomass. Fish oil (refined and without added antioxidants) was kindly provided by INVE (Baasrode, Belgium) and had an n-3 LC-PUFA content of 243 (±1) mg/g oil. No carotenoids were detected. N-3 LC-PUFA and carotenoid content were determined using the analyses described below. Both *Phaeodactylum* sp. biomass and fish oil were stored at −80 °C until further use.

#### 2.1.2. Chemicals

All solvents used for n-3 LC-PUFA, carotenoid and primary oxidation determination, and for lipid extraction were HPLC or GC-grade and were purchased from Roth (Karlsruhe, Germany) (chloroform, methanol, n-hexane ≥95%, acetonitrile) or VWR (Fontenay-sous-Bois, France) (toluene and ethyl acetate). Reagents were the following: xylenol orange disodium salt, iron(II) sulfate heptahydrate ≥99% (Sigma-Aldrich, Steinheim, Germany), barium chloride dihydrate p.a. (Chem-Lab NV, Zedelgem, Belgium), triphenylphosphine 99% (Sigma-Aldrich, Buchs, Switzerland), iron(III)chloride (Merck Schuchardt OHG, Hohenbrunn, Germany) and ammonium acetate (Merck KGaA, Darmstadt, Germany). Standards of fucoxanthin, diatoxanthin, and β-carotene for carotenoid analyses were purchased from DHI (Horsholm, Denmark) and standards for secondary oxidation analysis were acquired from Sigma-Aldrich (Steinheim, Germany) (2,4-heptadienal and (E)-2-pentenal) and Roth (Karlsruhe, Germany) (2-butenal).

### 2.2. Experimental Set-Up

The experimental set-up consisted of two parts, of which Figure 1 shows a schematic overview. ‘Part I’ investigated the oxidative stability of the following two systems: fish oil with and without added *Phaeodactylum* biomass. For ‘Part II’ six different systems obtained after mixing a *Phaeodactylum* total lipid extract with fish oil in different proportions were considered. Details of both experimental set-ups are discussed below.

#### 2.2.1. Part I

The fish oil was mixed with 2.5% (*w*/*w*) *Phaeodactylum* biomass and stirred for 5 min (room temperature, 300 rpm, IKA RO10 stirrer) to obtain the ‘Fish oil + *Phaeodactylum* biomass’ system. The concentration was chosen in line with former research adding microalgal biomass to bulk oil [19,20,21,22,23,24]. To obtain a correct impression of the influence of *Phaeodactylum* biomass addition on the oxidative stability, also a ‘Fish oil’ (control) system was considered, obtained by stirring pure fish oil for 5 min without *Phaeodactylum* biomass addition. The systems were characterized in terms of n-3 LC-PUFA content and carotenoid profile according to the analyses described below. Analyses of the ‘Fish oil + *Phaeodactylum* biomass’ system were performed on the oil phase obtained after sedimentation of the solid parts by a centrifugation step of 5 min at 1000 rpm (SL 16 Centrifuge, Thermo Scientific). To determine the oxidative stability a storage experiment of 3 weeks was set up. Both systems were transferred to amber screw-capped 20 mL vials in quantities of 1 g/vial, and while doing so, the systems were stirred with a magnetic stirrer to ensure a homogenous distribution. Vials were incubated (Memmert IN160 incubator, MLS, Menen, Belgium) at 37 °C (a temperature that accelerates the oxidation without changing its mechanisms [32]) and transferred to −80 °C until further analysis after seven different time points (day 0, 1, 3, 5, 8, 14, 21). For each time point, primary (peroxide value (PV)) and secondary (volatile compounds) oxidation products were determined in duplicate according to the methods described below, starting from independent vials. For the PV analysis of the ‘Fish oil + *Phaeodactylum* biomass’ system the oil phase (separated by centrifugation) was used again. A storage experiment analyzing the evolution of both primary and secondary oxidation products throughout storage was opted for to obtain a realistic indication of the oxidative stability and therefore the underlying antioxidant potential, rather than using quick antioxidant assays that only give a method-dependent indication [33].

#### 2.2.2. Part II

##### *Phaeodactylum* Total Lipid Extract

The second part of the experimental set-up started with obtaining a total lipid extract from the *Phaeodactylum* biomass. For this, a large-scale chloroform-methanol (1:1) extraction was used based on the method described by Ryckebosch et al. [34]. The following steps were performed in eightfold: 22.5 mL of the solvent was added to 3 g *Phaeodactylum* biomass and stirred for 90 min, after centrifugation (10 min, 2000 rpm) the solvent phase was separated and the same procedure was repeated. both combined solvent phases were washed with 11.25 mL water and after vigorous shaking (2 min), centrifugation (10 min, 2000 rpm), and removal of the upper water phase, the solvent phase was dried over sodium sulfate using a Whatman 1 filter that was washed with chloroform-methanol (1:1) afterward. All eight filtered extracts were pooled to obtain one homogenous extract and the solvent was removed by rotary evaporation under reduced pressure at 40 °C.

##### Preparation and Characterization of the Systems

Starting from the *Phaeodactylum* total lipid extract and the fish oil, six different systems were prepared by mixing them in different proportions. They were denoted as ‘Pure *Phaeodactylum* total lipid extract’, ‘Mixture 79/21’, ‘Mixture 49/51’, ‘Mixture 6/94’, ‘Mixture 1/99’ and ‘Pure Fish oil’. Table 1 shows an overview of the ratios used (in weight%). These ratios were chosen in such a way that different orders of magnitude of the amount of carotenoids relative to the amount of n-3 LC-PUFA could be evaluated. To ensure homogenous mixing, both phases were dissolved in chloroform-methanol (1:1) before combining them. All systems were characterized in terms of n-3 LC-PUFA content and carotenoid profile. For this, part of the mixture was taken aside, flushed dry with nitrogen, and analyzed (in triplicate) according to the methods described below.

##### Oxidative Stability (Storage Experiment)

The oxidative stability of the different systems was followed throughout a storage experiment of 10 weeks at 37 °C, within triplicate analysis of primary and secondary oxidation products (according to the methods described below) on the following time points: day 0, 4, 8, 12, 20, 28, 42, 56, and 70. The systems that already showed clear oxidation in the first 4 weeks were only analyzed until day 28. The systems were incubated in quantities of 10 mg in (independent) amber screw-capped 20 mL vials, obtained after distributing the different mixed systems (in solvent) and dry flushing with nitrogen. After incubation and until analysis, the vials were transferred to −80 °C.

### 2.3. Analyses

#### 2.3.1. N-3 LC-PUFA Content

The n-3 LC-PUFA content was determined as a part of the total fatty acid profile determination with gas chromatography (GC) after methyl transesterification according to the method described by Gheysen et al. [17] To enable quantitative analysis a known amount of lauric acid (C12:0) was added as an internal standard before derivatization.

#### 2.3.2. Carotenoids

Carotenoids were determined with high-pressure liquid chromatography (HPLC) with photodiode array detection (PAD) according to the method described by Gheysen et al. [18] with the difference that no extraction was needed. Instead, prior to HPLC analysis, a 10 mg lipid sample was dissolved in 10 mL acetone-methanol (7:3), filtered through a PVDF syringe filter (4 mm, 0.2 µm), and diluted 1:3. Identification and quantification were based on external calibration curves with fucoxanthin, diatoxanthin, and β-carotene standards.

#### 2.3.3. Primary Oxidation Products: Peroxide Value

The primary oxidation products were determined in terms of peroxide value (PV) using the spectrophotometric ferrous oxidation xylenol orange (FOX) method, including a triphenylphosphine (TPP) correction for possible interferences, described by Gheysen et al. [35]. The results were expressed as delta values, representing the relative increase or decrease compared to day zero, to focus on the trend of oxidation during storage.

#### 2.3.4. Secondary Oxidation Products: Volatile Compounds

The degree of secondary oxidation was determined by analysis of specific volatile compounds with headspace solid-phase microextraction Arrow gas chromatography–mass spectrometry (HS-SPME Arrow GC-MS) (GC Trace 1300 + ISQ 7000 single quadrupole MS, Thermo Scientific, Interscience, Louvain-la-Neuve, Belgium). After incubation (60 °C, 15 min), the volatile compounds were extracted (60 °C, 15 min) on a PAL DVB/CWR/PDMS SPME Arrow 1.1 mm fiber (CTC Analytics, Zwingen, Switzerland). A subsequent 2 min injection with a 1/50 split ratio and a desorption temperature of 250 °C was performed on a Rxi-1ms column (50 m length, 0.20 mm ID, 0.33 µm film) (Restek, Bellefonte, PA, USA) after which a time-temperature program from 40 °C to 250 °C was followed (5 min at 40 °C, increase of 5 °C/min, 5 min at 250 °C). The transfer lines worked at 280 °C. The MS operated with electron ionization at 70 eV and the scanned mass units ranged from 40 to 250 amu. For monitoring and data processing Chromelon 7 Chromatography Data System software (Thermo Scientific, Interscience, Louvain-la-Neuve, Belgium) was used. Several volatile compounds can be considered as a marker for secondary oxidation, originating from n-3 LC-PUFA hydroperoxides formed during primary oxidation [5,36,37,38,39]. However, only the compounds with a solid identification for all time points when using the NIST MS Search 2.3 spectral library, and of which a standard was available to confirm the retention time, were further reported. These are as follows: (E,E)-2,4-heptadienal, (E)-2-pentenal and 2-butenal. Calibration plots were set up by spiking these standards in the matrix of interest. The as such obtained semi-quantitative data (expressed in µg/mg) were again expressed as delta values, representing the relative increase or decrease compared to day zero, to focus on the oxidation trend.

### 2.4. Statistical Analysis

For the results of both primary and secondary oxidation, the impact of storage time was statistically evaluated for all systems with JMP Pro 15.1.0 software (SAS Institute Inc., Cary, NC, USA). One-way analysis of variance (ANOVA) with a significance level of 0.05 was used with a post hoc Tukey test in case of significance to determine the time points showing relevant increases or decreases.

## 3. Results and Discussion

### 3.1. Phaeodactylum Biomass Addition (Part I)

#### 3.1.1. Primary Oxidation

Figure 2 shows the primary oxidation results of the fish oil with and without *Phaeodactylum* biomass addition. Both systems show a significant increase towards high ΔPV values, with the highest ones being observed after 14 days of storage, as follows: 250 (±37) and 397 (±8) meq hydroperoxides/kg lipids with and without *Phaeodactylum* biomass addition, respectively. The order of magnitude of these values was comparable to former research investigating primary oxidation during the storage of bulk fish oil [16,40]. After 21 days, a decrease could be observed. This characteristic pattern of an increase followed by a decrease in primary oxidation products indicates strong lipid oxidation [5], and this is for fish oil as well as for fish oil with a 2.5% *Phaeodactylum* biomass addition. Although *Phaeodactylum* biomass addition gave rise to lower ΔPV values for most of the time points (between 8 and 161 meq hydroperoxides/kg lipids lower), it did not inhibit the lipid oxidation given the increase and decrease in primary oxidation products at the same time points. A possible reason for the lower ΔPV values (despite the increase and decrease at similar time points) could be found in the polarity increase in hydroperoxidized lipids. As described by Junqueira et al. [41], who investigated molecular organization in hydroperoxidized lipid bilayers, the −OOH moiety is preferentially located towards the more polar regions. Given the sampling method of the ‘Fish oil + *Phaeodactylum* biomass’ system in which the oil phase was separated from the solid *Phaeodactylum* parts by centrifugation prior to PV determination, this could have caused an underestimation of hydroperoxides by an accumulation away from the apolar bulk oil phase (towards the *Phaeodactylum* solid parts).

#### 3.1.2. Secondary Oxidation

Figure 3 shows the evolution of (E,E)-2,4-Heptadienal, (E)-2-Pentenal, and (Z)-2-Butenal as a measure for secondary n-3 LC-PUFA oxidation. For both systems, these three compounds showed a clear increase of 0.06 µg/mg, 0.05 µg/mg, and 0.03 µg/mg throughout storage, respectively. The fact that a clearer (and significant) increase was observed towards the end of the storage experiment (after between 14 and 21 days), while the primary oxidation products showed a sooner (significant) increase (during the first storage week), is logical as these compounds are formed through hydroperoxide decomposition [36,38,39]. Other than for the primary oxidation results, the fish oil systems with *Phaeodactylum* biomass addition (roughly) showed an equally large (but certainly not lower) increase compared to the pure fish oil systems. Apart from making the above-mentioned hypothesis of the hydroperoxide underestimation more probable, this again indicates clear lipid oxidation independent of 2.5% *Phaeodactylum* biomass addition.

#### 3.1.3. Antioxidative Capability

Based on the results of primary and secondary oxidation, it is clear that the *Phaeodactylum* biomass addition did not exert an antioxidative effect. In other words, *Phaeodactylum* biomass was not capable of inhibiting fish oil oxidation. Former studies that added microalgal biomass to bulk oil did, however, report an enhancement in oxidative stability [19,20,21,22,23,24]. Nonetheless, it has to be remarked that Lee et al. [19], who investigated the addition of *Nannochloropsis* biomass to soybean oil, based these findings on a Rancimat test operating at temperatures of 120–150 °C far from real storage conditions, while single PV values after 15 h (without further storage) were even shown to be slightly higher (ca. 2 meq hydroperoxides/kg oil) for the oil with added *Nannochloropsis* biomass. Moreover, the results of the studies of Alavi and Golmakani [20,22] require nuance. Although microalgal biomass addition (in this case, *Chlorella* or *Arthrospira*) to virgin olive oil did lead to lower PV values (of ca. 40–100 meq hydroperoxides/kg oil lower), this was only the case for the last time point of the 42-day storage experiment that (compared to the earlier time points) still showed a clear increase of ca. 50–100 meq hydroperoxides/kg oil and therefore an oxidation onset for oils both with and without microalgal biomass. Moreover, it has to be kept in mind that, other than in the current research, systems without n-3 LC-PUFA were considered, and also the methods of analyzing lipid oxidation showed some differences since a storage temperature of 60 °C and a less specific analysis method for secondary oxidation (anisidine value) were used. This was also the case in the study of Morsy et al. [24], which investigated the effect of adding both whole-cell *Arthrospira* and *Arthrospira* nanoparticles (after ball-milling) to olive oil, although, in this study, lower PV values (4–58 meq hydroperoxides/kg oil lower) without a clear oxidation onset were observed for all time points (weekly analysis for 7 weeks), making their statement of *Arthrospira* exerting an antioxidative action more powerful. Since in all of these papers, the antioxidative effect is explained by the release of antioxidative compounds from the microalgal biomass into the oil, it raises the question of whether this sufficiently occurred in the current research or whether the reason for the absence of oxidation inhibition lies in the fact that no antioxidants were released from the *Phaeodactylum* biomass. With former research suggesting carotenoids as the most important antioxidants in *Phaeodactylum* [18,29,31], their presence in the fish oil after mixing with *Phaeodactylum* biomass was investigated and is discussed in the next section.

#### 3.1.4. Carotenoid Content

Table 2 shows the results of the carotenoid characterization of the oil phase of the *Phaeodactylum* biomass-enriched fish oil system. It concerns the content of carotenoids after preparing the system (i.e., at the start of the storage experiment). Just like in the *Phaeodactylum* biomass, fucoxanthin was the most abundant carotenoid, followed by diatoxanthin and β-carotene. The fact that carotenoids were detected in the oil showed that during stirring, a certain carotenoid transfer from the *Phaeodactylum* biomass to the fish oil occurred, as no carotenoids were present in the pure fish oil. Based on the amount of fucoxanthin in the *Phaeodactylum* biomass (5.31 (±0.52) mg/g dry biomass), the amount of fucoxanthin in the oil phase in the case of full transfer would have been 133 (±13) µg/g, indicating that ca. 12 (±3)% of the fucoxanthin migrated. Although it could therefore be stated that the absence of antioxidant migration was not the reason for the lack of oxidation inhibition, it is still possible that the amount did not suffice for an antioxidative effect. Limon et al. [27] reported carotenoid contents of ca. 20 µg β-carotene and ca. 6 µg lutein per g oil in their study that did describe an antioxidative effect of *Scenedesmus* extract addition to virgin olive oil. These contents are comparable to the amount of fucoxanthin detected in the current research. Although the differences in set-up regarding raw materials (not containing n-3 LC-PUFA) and analysis methods (Rancimat) could again be remarked as possible reasons for different findings, the question remains why the microalgal antioxidants in the current research were not capable of keeping the n-3 LC-PUFA rich fish oil oxidatively stable while their role in doing so has been suggested for the n-3 LC-PUFA rich microalgal oil or biomass itself [16,18].

In trying to address this question, apart from only considering the carotenoid content itself, the amount relative to the amount of n-3 LC-PUFA (the compounds causing oxidative susceptibility) was also examined. Table 2 shows the ratio of carotenoid to n-3 LC-PUFA content of the oil phase, with the fucoxanthin to n-3 LC-PUFA ratio being 0.00007 (±0.00002). If this is compared with the pure *Phaeodactylum* biomass having a fucoxanthin/n-3 LC-PUFA ratio calculated to be 0.27 (±0.03), the big difference in the order of magnitude is clear. Despite antioxidants being defined as compounds that delay or prevent oxidation when present at low concentrations compared with those of the oxidizable substrate [10], the observed concentrations in the oil phase may simply be too low. Investigating the amount that would suffice for stabilizing the fish oil could give more insight into the capacity of *Phaeodactylum* as an antioxidant and forms the basis of the second part of the experimental set-up. The potential of *Phaeodactylum* biomass addition will be returned at the end of Part II, taking into account its results.

### 3.2. Phaeodactylum Total Lipid Extract–Fish Oil Mixtures (Part II)

#### 3.2.1. Characterization (Fucoxanthin/n-3 LC-PUFA Ratio)

Building further on the amount of fucoxanthin relative to the amount of n-3 LC-PUFA, Table 3 gives the n-3 LC-PUFA content (in mg/g), fucoxanthin content (in mg/g), and fucoxanthin/n-3 LC-PUFA ratio (in *w*/*w*) of the different mixtures of *Phaeodactylum* total lipid extract and fish oil. Fucoxanthin was further highlighted in this because of its abundance and its importance in former research [18,29]. While other antioxidants might also play their role [31], fucoxanthin was thus considered the major antioxidant. Information on the full fatty acid and carotenoid composition of both *Phaeodactylum* total lipid extract and pure fish oil can be found in Appendix B.

Since the fish oil has a higher n-3 LC-PUFA content than the *Phaeodactylum* total lipid extract (between 227 and 117 mg/g, respectively), it should not surprise that the n-3 LC-PUFA content increases as relatively more fish oil were used. The fucoxanthin content, on the other hand, decreases by blending more fish oil due to its absence of carotenoids. Therefore, also the fucoxanthin/n-3 LC-PUFA ratio decreases, from which it is clear that mixing the *Phaeodactylum* total lipid extract and fish oil in the amounts that were described in Table 1 led to different orders of magnitude of the fucoxanthin/n-3 LC-PUFA ratio. This formed an interesting starting point to investigate the oxidative stability of the different systems.

#### 3.2.2. Primary Oxidation

Figure 4 shows the primary oxidation results of the six different systems. A clear separation in ΔPV progression between ‘Pure *Phaeodactylum* total lipid extract’, ‘Mixture 79/21’ and ‘Mixture 49/51’ on the one hand, and ‘Mixture 6/94’, ‘Mixture 1/99’ and ‘Pure Fish oil’ on the other hand was observed. The first three showed a rather small, insignificant increase in ΔPV during storage, indicating that primary oxidation was only minor or absent. After 10 weeks, increases of 52 (±33), 19 (±4), and 21 (±13) meq hydroperoxides/kg lipids were obtained, respectively. For the last three systems, much higher ΔPV values were detected, with a significant increase towards a maximum already after 4 days of storage, followed by a significant decrease after 8 days. Since a clear and early primary oxidation peak was observed, respectively, with ΔPV maxima of 341 (±108), 402 (±79), and 491 (±46) meq hydroperoxides/kg lipids for ‘Mixture 6/94’, ‘Mixture 1/99’, and ‘Pure Fish oil’, these systems were only analyzed during 4 weeks. Based on these results, it could be stated that regarding primary oxidation, ‘Mixture 79/21’ and ‘Mixture 49/51’ were in line with the pure *Phaeodactylum* extract, of which the tendency not to lead to high ΔPV values was comparable with former research following primary oxidation during storage of *Phaeodactylum* hexane-isopropanol (3:2) extracts [16]. ‘Mixture 6/94’ and ‘Mixture 1/99’, on the other hand, were in line with the pure fish oil, for which a clear primary oxidation peak showed its high oxidative susceptibility. Comparing the primary oxidation results of the pure fish oil with those of Part I (3.1.1.), it was clear that the ΔPV peak occurred sooner. This could be attributed to the fact that in Part II amounts of 10 mg/vial instead of 1 g/vial were used, leading to a higher oxygen to bulk oil availability, explaining the accelerated oxidation processes [42]. Ryckebosch et al. [16] also observed a primary oxidation peak after 7 days (as the first time point) for refined fish oil being stored at 37 °C in amounts of 10 mg/vial with a weekly PV analysis.

#### 3.2.3. Secondary Oxidation

The clear separation regarding the course of primary oxidation between ‘Pure *Phaeodactylum* total lipid extract’, ‘Mixture 79/21’ and ‘Mixture 49/51’, on the one hand, and ‘Mixture 6/94’, ‘Mixture 1/99’ and ‘Pure Fish oil’ on the other hand, was also visible for the secondary oxidation products, as can be seen in Figure 5 showing the progression of (E,E)-2,4-Heptadienal, (E)-2-Pentenal and (Z)-2-Butenal as secondary n-3 LC-PUFA oxidation indicators. For all three compounds, the ‘Mixture 6/94’, ‘Mixture 1/99’, and ‘Pure Fish oil’ systems showed a clear, significant increase (of respectively ca. 0.02 µg/mg, ca. 0.3 µg/mg and ca. 0.15 µg/mg) already after 4 days of storage. Other than in Part I, this significant increase in secondary oxidation products did not appear later than that of the primary oxidation products. Again, the relatively higher amount of oxygen as compared with the amount of bulk lipids could have led to accelerated oxidation, making the differentiation between the onset of primary and secondary oxidation less clear. The fact that also for the secondary oxidation products a clear, significant increase was already observed after the first time point indicated thorough lipid oxidation for all three systems. The ‘Pure *Phaeodactylum* total lipid extract’, ‘Mixture 79/21’, and ‘Mixture 49/51’ systems, on the other hand, showed a flat course throughout 10 weeks of storage for all three compounds. This indicated the absence of clear secondary lipid oxidation, which is in line with the *Phaeodactylum* hexane-isopropanol (3:2) extract in the research of Ryckebosch et al. [16], and which also corresponds with the above-described small increase in primary oxidation products for these three systems. However, other than for the primary oxidation results, Tukey analysis showed that for the (E,E)-2,4-heptadienal and (E)-2-pentenal content, a small but significant decrease (of ca. 0.002 µg/mg) occurred after the first time point. Similarly, as described for *Nannochloropsis* biomass by Gheysen et al. [18], a reason for this reduction could be that volatile organic compounds (VOCs), previously formed in vivo in *Phaeodactylum* and co-extracted in the total lipid extract, were already present at the start of the storage experiment and decomposed during storage. Although in the literature, (E,E)-2,4-heptadienal and (E)-2-pentenal are not specifically mentioned as known *Phaeodactylum* VOCs [43], Pohnert and Boland [44] described the in vivo formation of lipid oxidation derived unsaturated aldehydes (encompassing (E,E)-2,4-heptadienal and (E)-2-pentenal) as a defense mechanism for diatoms, which would explain their presence. Anyhow, it is clear that this phenomenon (although significant) was only minor compared to the clear secondary lipid oxidation in the ‘Mixture 6/94’, ‘Mixture 1/99’, and ‘Pure Fish oil’ systems.

#### 3.2.4. Oxidative Stability and Antioxidative Capacity

The results of both primary (Section 3.2.2) and secondary oxidation (Section 3.2.3) showed a clear separation in oxidative stability between the different systems. ‘Mixture 79/21’ and ‘Mixture 49/51’ (containing respectively 78.6% and 49.2% (*w*/*w*) of *Phaeodactylum* total lipid extract) have been shown to be equally stable as the pure *Phaeodactylum* extract despite the presence of respectively 21.4% and 50.8% (*w*/*w*) of oxidatively susceptible fish oil. On the other hand, ‘Mixture 6/94’ and ‘Mixture 1/99’ (containing respectively 93.6% and 99.3% (*w*/*w*) of fish oil) have been shown to be equally unstable as the pure fish oil despite the presence of 6.4% and 0.7% (*w*/*w*), respectively, of *Phaeodactylum* total lipid extract. This clearly showed that apart from the presence of lipophilic antioxidants from the *Phaeodactylum* total lipid extract, of which the role in keeping *Phaeodactylum* lipids stable was already suggested in former research [16,18,29,31] and could again be observed in the oxidative stability of the (n-3 LC-PUFA rich) *Phaeodactylum* total lipid extract of the current research, their amount also plays a role in determining lipid oxidation to be either absent or far advanced.

Revising the fucoxanthin/n-3 LC-PUFA ratio of the different systems (Section 3.2.1.), it could be stated that systems with a fucoxanthin/n-3 LC-PUFA ratio ≥ 0.101 were shown to be oxidatively stable, whereas systems with a ratio ≤ 0.0078 led to strong lipid oxidation. In this regard, the fact that 2.5% *Phaeodactylum* biomass addition to the fish oil did not show inhibition of oxidation (Part I) is not surprising since this gave rise to a fucoxanthin/n-3 LC-PUFA ratio of only 0.00007 (±0.00002) in the oil phase. These findings, therefore, confirmed the hypothesis of antioxidant concentrations being too low as compared to n-3 LC-PUFA (as the oxidizable substrate) to have an antioxidative effect, although they were present in similar absolute concentrations as in former research where an antioxidative effect had been described [27]. Interesting to remark is that regarding the fucoxanthin migration to the oil phase in Part I, even a full transfer would not have led to a fucoxanthin/n-3 LC-PUFA ratio associated with oxidative stability. Therefore, no effort should be made in trying to improve antioxidant migration by additional treatments in trying to obtain an antioxidative effect of 2.5% *Phaeodactylum* biomass addition to fish oil. Moreover, significantly increasing the biomass concentration would also not lead to the fucoxanthin/n-3 LC-PUFA ratios of interest. An extreme, tenfold increase to 25% *Phaeodactylum* biomass, for example, would, whether or not feasible, give rise to a fucoxanthin/n-3 LC-PUFA ratio of 0.0007 (±0.0002) maximally (assuming a similar migration). Unlike other former research adding microalgal biomass as an antioxidant to bulk oil, of which the antioxidative effects of Morsy et al. [24] (*Arthrospira* addition to olive oil) were the most convincing, it is clear that *Phaeodactylum* biomass addition to stabilizing fish oil thus has no potential.

## 4. Conclusions

In conclusion, it can be stated that *Phaeodactylum* biomass (2.5% (*w*/*w*)) was not capable of inhibiting n-3 LC-PUFA oxidation in fish oil since the progression of both primary and secondary oxidation products showed strong lipid oxidation with and without biomass addition. *Phaeodactylum* carotenoids were, however, shown to be present in the fish oil phase after mixing, but these did not suffice to induce oxidative stability since, also, apart from their presence, their relative amounts as compared to the n-3 LC-PUFA content were of importance. This was clearly shown in the second experimental set-up in which the oxidative stability of *Phaeodactylum* total lipid extract and fish oil was followed after being mixed in different proportions, leading to different carotenoid to n-3 LC-PUFA ratios. Systems with a fucoxanthin/n-3 LC-PUFA ratio ≥ 0.101 were associated with oxidative stability, whereas systems with a ratio ≤ 0.0078 showed extreme lipid oxidation. Apart from explaining the absence of oxidation inhibition in the case of 2.5% (*w*/*w*) *Phaeodactylum* biomass addition, this also showed that tenfold bigger amounts of biomass (whether practically feasible or not) would not be capable of obtaining this. Investigating how generalizable these findings are for other fucoxanthin-rich microalgal species (with a focus on the importance of this ratio) would be an interesting question for further research.

## Figures and Tables

**Figure 1 foods-11-01461-f001:**
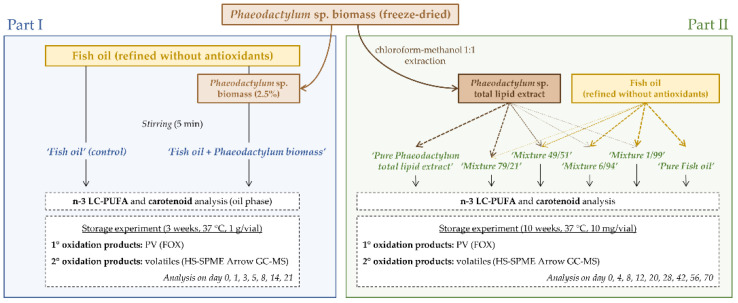
Experimental set-up.

**Figure 2 foods-11-01461-f002:**
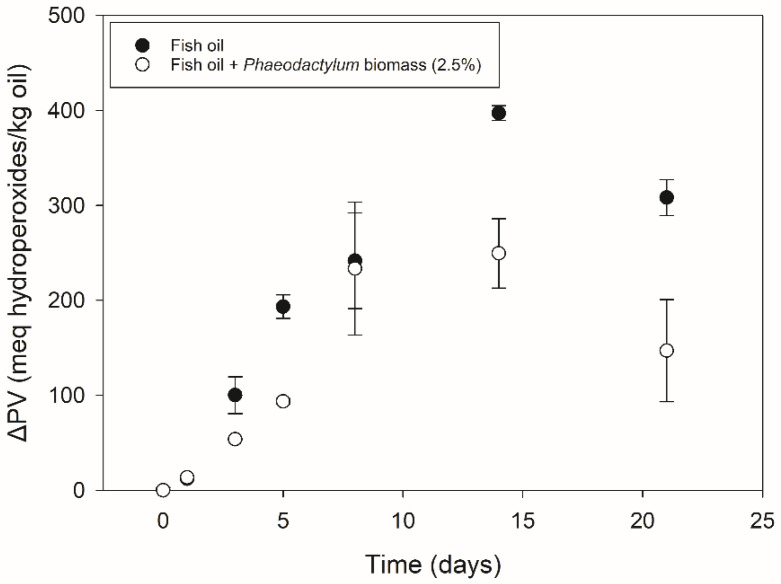
Primary oxidation of fish oil with (white symbols) and without (black symbols) 2.5% *Phaeodactylum* biomass addition during 21 days of storage at 37 °C. The peroxide values (mean ± sd, *n* = 2) are expressed as delta values, representing the relative increase or decrease compared to day zero. Information on statistical significance can be found in Appendix A.

**Figure 3 foods-11-01461-f003:**
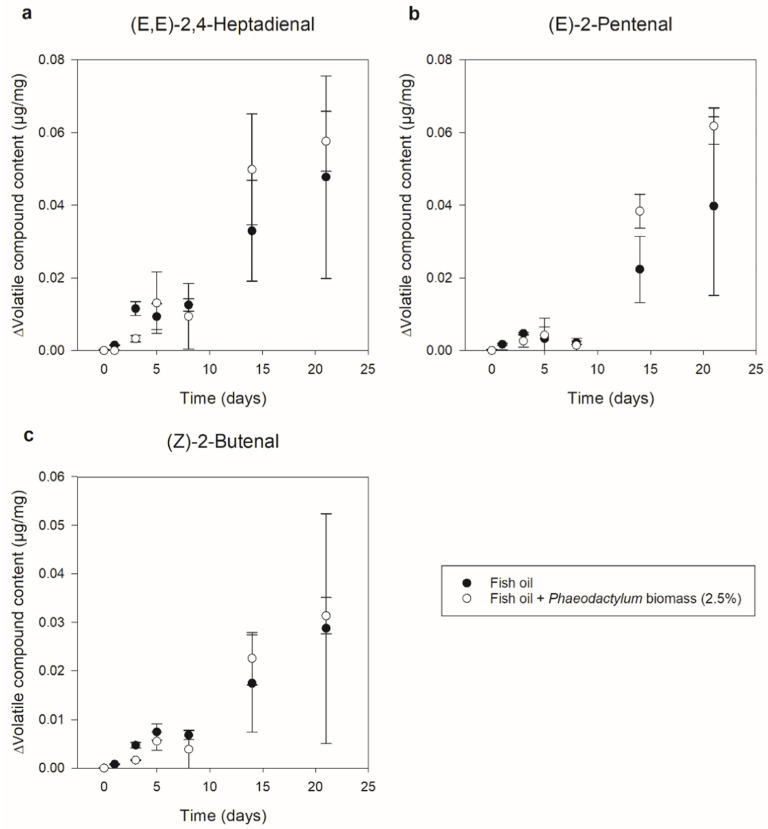
Secondary oxidation of fish oil with (white symbols) and without (black symbols) 2.5% *Phaeodactylum* biomass addition during 21 days of storage at 37 °C. The content of volatile compounds originating from lipid oxidation [(E,E)-2,4-Heptadienal (**a**), (E)-2-Pentenal (**b**) and (Z)-2-Butenal (**c**)] in µg/mg (mean ± sd, *n* = 2) is expressed in delta values, representing the relative increase or decrease compared to day zero. Information on statistical significance can be found in Appendix A.

**Figure 4 foods-11-01461-f004:**
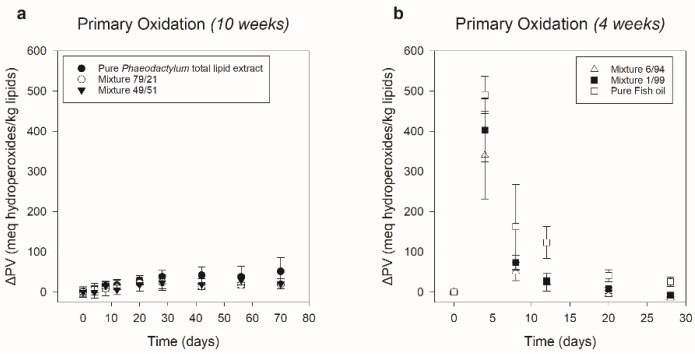
Primary oxidation of six different ‘*Phaeodactylum* total lipid extract–fish oil’ systems during storage at 37 °C. ‘Pure *Phaeodactylum* total lipid extract’, ‘Mixture 79/21’ and ‘Mixture 49/51’ (respectively possessing 100.0%, 78.6% and 49.2% (*w*/*w*) of *Phaeodactylum* total lipid extract) were stored for 10 weeks (**a**). ‘Mixture 6/94’, ‘Mixture 1/99’, and ‘Pure Fish oil’ (respectively possessing 6.4%, 0.7% and 0.0% (*w*/*w*) of *Phaeodactylum* total lipid extract) were stored for 4 weeks (**b**). The peroxide values (mean ± sd, *n* = 3) are expressed as delta values, representing the relative increase or decrease compared to day zero. Information on statistical significance can be found in Appendix A.

**Figure 5 foods-11-01461-f005:**
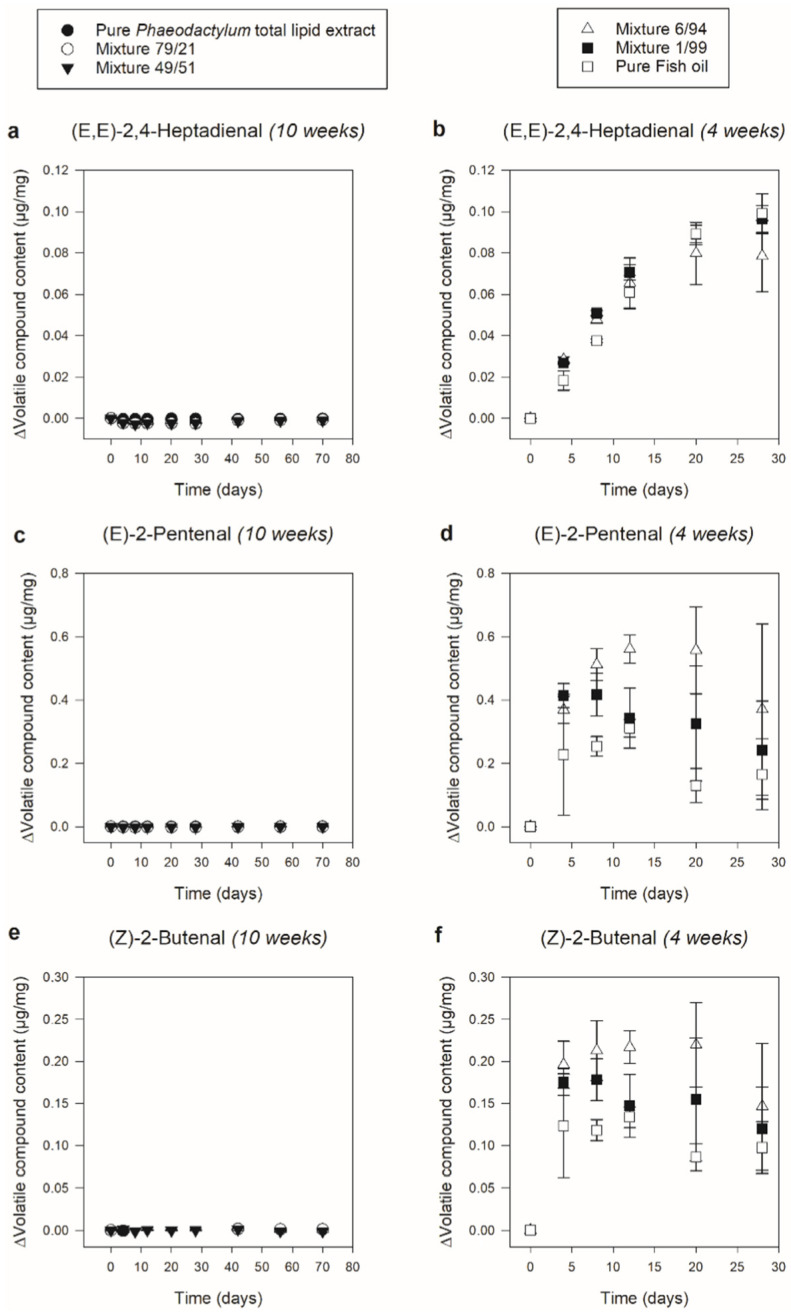
Secondary oxidation of six different ‘*Phaeodactylum* total lipid extract–fish oil’ systems during storage at 37 °C. ‘Pure *Phaeodactylum* total lipid extract’, ‘Mixture 79/21’, and ‘Mixture 49/51’ (respectively possessing 100.0%, 78.6% and 49.2% (*w*/*w*) of *Phaeodactylum* total lipid extract) were stored for 10 weeks (left). ‘Mixture 6/94’, ‘Mixture 1/99’, and ‘Pure Fish oil’ (respectively possessing 6.4%, 0.7% and 0.0% (*w*/*w*) of *Phaeodactylum* total lipid extract) were stored for 4 weeks (right). The content of volatile compounds originating from lipid oxidation [(E,E)-2,4-Heptadienal (**a**,**b**), (E)-2-Pentenal (**c**,**d**), (Z)-2-Butenal (**e**,**f**)] in µg/mg (mean ± sd, *n* = 3) is expressed in delta values, representing the relative increase or decrease compared to day zero. Information on statistical significance can be found in Appendix A.

**Table 1 foods-11-01461-t001:** Ratios (in weight%) of the *Phaeodactylum* total lipid extract and fish oil used for preparing the different systems of Part II of the experimental set-up.

System	*Phaeodactylum* Total Lipid Extract (Weight%)	Fish Oil (Weight%)
Pure *Phaeodactylum* total lipid extract	100.0	0.0
Mixture 79/21	78.6	21.4
Mixture 49/51	49.2	50.8
Mixture 6/94	6.4	93.6
Mixture 1/99	0.7	99.3
Pure Fish oil	0.0	100.0

**Table 2 foods-11-01461-t002:** Oil phase carotenoid content of the ‘Fish oil + *Phaeodactylum* biomass’ system.

Carotenoid	Content (µg/g Oil)	Carotenoid/n-3 LC-PUFA Ratio (*w*/*w*)
Fucoxanthin	17 (±4)	0.00007 (±0.00002)
Diatoxanthin	4.4 (±0.7)	0.000018 (±0.000003)
β-carotene	<LOQ (=5)	<0.00002

**Table 3 foods-11-01461-t003:** Characterization of the different *Phaeodactylum* total lipid extract–fish oil mixtures in terms of n-3 LC-PUFA and fucoxanthin.

System	N-3 LC-PUFA (mg/g)	Fucoxanthin (mg/g)	Fucoxanthin/n-3 LC-PUFA Ratio (*w*/*w*)
Pure *Phaeodactylum* total lipid extract	117 (±1)	32.4 (±0.4)	0.277 (±0.003)
Mixture 79/21	145 (±3)	24.06 (±0.04)	0.1663 (±0.0003)
Mixture 49/51	165 (±4)	16.7 (±0.3)	0.101 (±0.002)
Mixture 6/94	222 (±3)	1.73 (±0.04)	0.0078 (±0.0002)
Mixture 1/99	232 (±3)	0.211 (±0.004)	0.00091 (±0.00002)
Pure Fish oil	227 (±6)	-	-

## Data Availability

Not applicable.

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
