# Peer review of "The Potential of Phaeodactylum as a Natural Source of Antioxidants for Fish Oil Stabilization"

_foods, 2022, doi:10.3390/foods11101461_

Round 1

Reviewer 1 Report

Dear Authors,

The article is interesting and provide an interesting view about the use of natural antioxidants in n-3-rich oils (susceptible to lipid oxidation). In general, the current document has a fair design, but more effort is still necessary to clarify important questions and considerations about the experiments. My specific comments are indicated below.

Lines 13-14: Please revise this sentence for clarity.

Line 15: Please revise this sentence.

Lines 22-23: The meaning of this sentence is not clear. Please revise it.

Line 28: Please delete the phrase “With 10% of the global fish production being committed to its manufacturing,”

Line 32-33: Please revise this sentence for clarity.

Line 36: Please replace “For this” by “Consequently”.

Line 88: Replace “going from pure Phaeodactylum extract to pure fish oil” by the actual proportions/concentrations of extract used in the experiment.

Table 1. The nomenclature of treatments is confusing. Treatments with the dilution of microalgae extract in fish oil and the opposite (fish oil in microalgae extract) are receiving the very similar names (Dilution 0.01 and Dilution 0.001 vs. Dilution 0.2 and Dilution 0.1, respectively). Please change the name of treatments.

Table 1. If the objective of the study is the evaluation of microalgae biomass/extract as natural antioxidant in fish oil, what is the reason that justify the treatments Dilution 0.2 and Dilution 0.1? In this case, the mix is mainly composed of microalgae extract and could be seen more like a blended oil than fish oil with natural extract.

Lines 190-191: The standard and its brand should be presented in section 2.1.2. Chemicals.

Line 226: One-way ANOVA is the appropriate test.

Why the antioxidant potential was not been evaluated? ORAC assay, for instance.

Results and discussion. Without proper indication of statistical results, it is not possible to interpret or discuss and results. Moreover, no conclusion can be given about the experiment. Please redo the statistical tests and indicate it the appropriate figures and tables.

Figure 2: Is there any statistical difference in this data? Is there significant interactions between variables? If so, indicate with letters/symbols.

Line 239: This sentence is incomplete. Please revise it.

Lines 241-242: Does statistical analysis support this statement?

Lines 242-249: Essentially, the hypothesis for this particular test is that microalgae do not have antioxidant activity and, therefore, could not have slowed the progression of lipid oxidation. Is it right?

Lines 329-331: The comparison is not supported by the results of the experiment. No data about the oxidative stability of microalgae biomass was obtained in this study.

Lines 331-334: This statement contradicts the proposed treatments Dilution 0.2 and Dilution 0.1 (78.6 and 49.2% weight) for Table 1.

Lines 334-336: Part 1 and 2 are essentially different. Biomass it is not the same as an extract. Stablishing a relation for concentration using different systems (biomass vs. extract) is not scientifically correct considering only the concentration. It is crucial that this section also discuss the role of physical barriers. The lyophilized biomass is expected to exert a barrier effect and limit the content between oil (external phase) and natural antioxidants (internal phase; within the microalgae).

Why was a higher concentration of biomass not evaluated? Changing from biomass to extract was not properly justified. This justification must be clarified in the paper.

What is the composition of Phaeodactylum total lipid extract?

Table 3. Treatments A-D?

Line 347: “Phaeodactylum” must be italicized. Please check the whole manuscript.

Lines 368-372: Without the composition of Phaeodactylum total lipid extract is not possible to understand what is happening and if the PV assay is relevant or not. Another issue is the different nomenclature between Tables 1 and 3.

Lines 409-410: This information (after proper statistical evaluation) must be indicated in the Figure 5.

Line 413: “in vivo” must be italicized. Please check the whole manuscript.

Lines 440-445: This explanation makes sense and support the general hypothesis of the study, but only with the microalgae extract is rich in unsaturated fatty acids. Is the extract rich in unsaturated fatty acids?

Line 446: Section 3.2.1.

Lines 457-466: These lines should be moved to the end of section 3.1.4. Carotenoid content.

What is the content of n-3 fatty acids in each treatment? It important to connect the proposed treatments with the further application in food where the proposed treatment from this study provide nutritionally relevant content for food enrichment.

Conclusion: It is important to indicate that the proposed use of microalgae extract was not efficient to slow lipid oxidation when used as natural antioxidant (Dilution 0.01 and Dilution 0.001). The blend of lipids would provide a more stable product. In the case of blend, the further application would probably require the encapsulation of the blend to prevent sensorial alterations in food. Please include a comment in the conclusion about these considerations.

Figures 3 and 4 are composed of multiple graphs. Each one should be indicated with a letter (3a, 3b, 3c, 4a, 4b…) in the figure and in the caption of the figure.

Reviewer 2 Report

Comment on “The potential of Phaeodactylum as a natural source of antioxidants for fish oil stabilization”

  1. Introduction about the chemical composition of Phaeodactylum sp. is needed.
  2. Line 121. Why the concentration was fixed at 2.5%? What is the effective concentration of Phaeodactylum biomass?
  3. Line 124. Stirring for 5 min. Please clarify why only 5 min was used? How about the temperature and speed during stirring?
  4. Line 131. Is 1 g each vial a reasonable figure?
  5. In both experiments, why the storage was performed at 37°C? Why not accelerated temperature?
  6. Oxidative stability of fish oil with commercial antioxidants or pure fucoxanthin should be compared.
  7. Line 143. Why chloroform-methanol was used? The toxicity of the solvent should be considered.

Round 2

Reviewer 2 Report

All points raised by reviewers were carefully addressed and answered point-by-point. So, it can be accepted.

Author Response

Thank you for the approval.